# Independent Responses of Photosynthesis and Plant Morphology to Alterations of PIF Proteins and Light-Dependent MicroRNA Contents in *Arabidopsis thaliana pif* Mutants Grown under Lights of Different Spectral Compositions

**DOI:** 10.3390/cells11243981

**Published:** 2022-12-09

**Authors:** Pavel Pashkovskiy, Vladimir Kreslavski, Alexandra Khudyakova, Elena S. Pojidaeva, Anatoliy Kosobryukhov, Vladimir Kuznetsov, Suleyman I. Allakhverdiev

**Affiliations:** 1K.A. Timiryazev Institute of Plant Physiology, Russian Academy of Sciences, Botanicheskaya Street 35, Moscow 127276, Russia; 2Institute of Basic Biological Problems, Russian Academy of Sciences, Institutskaya Street 2, Pushchino, Moscow 142290, Russia

**Keywords:** *A. thaliana pif* mutants, light of different spectral compositions, microRNA, phytochrome interacting factor, photosynthesis

## Abstract

The effects of the quality of light on the content of phytochrome interacting factors (PIFs) such as PIF3, PIF4 and PIF5, as well as the expression of various light-dependent microRNAs, in adult *Arabidopsis thaliana pif* mutant plants (*pif4*, *pif5*, *pif3pif5*, *pif4pif5*, *pif3pif4pif5*) were studied. We demonstrate that under blue light, the *pif4* mutant had maximal expression of most of the studied microRNAs (miR163, miR319, miR398, miR408, miR833) when the PIF4 protein in plants was reduced. This finding indicates that the PIF4 protein is involved in the downregulation of this group of microRNAs. This assumption is additionally confirmed by the fact that under the RL spectrum in *pif5* mutants, practically the same miRNAs decrease expression against the background of an increase in the amount of PIF4 protein. Unlike the WT and other mutants, the *pif4* mutant responded to the BL spectrum not only by activating the expression of light-dependent miRNAs, but also by a significant increase in the expression of transcription factors and key light signalling genes. These molecular reactions do not affect the activity of photosynthesis but may be involved in the formation of a light quality-dependent phenotype.

## 1. Introduction

Study of the mechanisms of light signal perception and transmission in plants has demonstrated the presence of a coordinated signalling network that affects many types and regulatory systems. A wide range of wavelengths of visible light are perceived by certain types of photoreceptors. In addition to photoreceptors, other molecules include transcription factors (TFs) (such as phytochrome interacting factor (PIF), short hypocotyl in white light (SHW), bZIP transcription factors—G-box-binding factor (GBF), squamosa promoter binding protein-like seven (SPL7), far-red elongated hypocotyl (FHY), bHLH transcription factor-HFR1), light signalling factors (CRY-interacting bHLH-CIB, E3 ubiquitin-protein ligase—COP, light-mediated development protein—DET and others), protein kinases and other regulatory molecules. Recently, increasing attention has focused on the role of miRNAs in light signalling [1,2,3]. MicroRNAs are important regulators of plant ontogenetic development under normal and stress conditions. Light can also affect MIR gene transcription, miRNA biogenesis and the activity of the RNA-induced gene silencing complex (RISC), thereby controlling not only miRNA accumulation but also their biological functions. MicroRNAs can be regulated by TFs involved in plant responses to light of different spectra. MIR genes contain cis-elements in their promoter regions, which bind certain light-dependent TFs such as PIF4 [4]. It is known that the interaction of PIF4 with the photoactivated form of PHYB regulates a subgroup of downstream TFs by binding to the promoters of these genes [5,6]. The TF PIF4 functions at both the transcriptional and posttranscriptional levels, regulating miRNA biogenesis and binding directly to MIR gene promoters.

Despite the large amount of data on the role of microRNAs in light signalling, the “regulatory potential” of microRNAs has not yet been studied sufficiently. Sorin and co-authors studied the relationship between auxin and light signalling through Argonaute 1 (AGO1) proteins, which are catalytic components of RISC. It was found that the AGO1 protein, which is important for the expression of microRNA exonuclease activity, is involved in the activation of light-induced auxin signalling [7]. Since light affects microRNA processing, it is logical to assume that photoreceptors may be involved in this regulation. For example, it has been shown in *A. thaliana* seedlings that the most differentially expressed miRNAs under the red light (RL) spectrum are miR160, miR163, miR319, miR394, miR779, miR851, miR854 and miR2111 [4]. In *A. thaliana pif4* mutants, the levels of 22 miRNAs changed under RL exposure. PIF4 increases the intensity of the expression of genes encoding miR156/157, miR160, miR165/166, miR167, miR170/171,and miR394 and reduces the expression of the miR172 and miR319 genes [4]. In *Oryza sativa phyb* mutants, 32 microRNAs have been found that are able to regulate approximately 70 different TFs, which additionally indicates the involvement of microRNAs in PHYB-mediated light signalling [8]. This is supported by the fact that light-dependent miRNAs are involved in the plant photomorphogenesis of *A. thaliana phy* mutants under red light (RL) and blue light (BL) spectra [3], and red light photoreceptors are involved in miRNA biogenesis. All this indicates the presence of a relationship between microRNA biogenesis and RL signalling through PIF-dependent regulation of transcription and microRNA processing. However, PIF4 and PIF5 act as regulators of BL receptor phototropin 1 (PHOT1) [9]. Exposure to BL and UV-B can affect the expression of the miR166, miR167, miR393 and miR396 families, which are mainly associated with auxin signalling [10,11,12,13]. UV-A irradiation of plants reduced miR156/157 expression, which correlated with activation of their target TFs SPL9 and SPL15, which are involved in the regulation of many physiological processes, including seed germination and flowering [2,14,15,16,17,18]. Under the BL spectrum, another photoreceptor, CRY1, is able to directly interact with PIF4 and suppress its interaction with PHYB, suggesting coordination between BL (CRY1) and RL (PHYB) signalling [19].

As mentioned above, PIF4 can regulate the expression of light-dependent miRNAs subjected to RL. In addition, PIFs are also able to interact with blue light receptors, such as CRY and PHOT [9,19], but how this interaction leads to the expression of light-dependent microRNAs is poorly understood. Moreover, most of the experiments in this area have been performed on *A. thaliana* seedlings [4,5]; however, the question of how PIFs are involved in microRNA light regulation in adult plants remains open. The solution to this issue is relevant since it is known that the same molecules at different stages of ontogeny can perform different biological functions [20].

Given these considerations, it is extremely important to study the ability of PIFs to regulate the expression of RNAs, TF genes and key genes for light signalling and photosynthetic activity under BL and RL spectra. This allowed us to obtain new data on the implementation of light signalling in *pif* mutants under different light spectra in connection with plant photosynthetic activity and morphology.

## 2. Materials and Methods

### 2.1. Plant Materials and Experimental Design

Wild-type and mutant *A. thaliana* plants (CS66043 (*pif4*), CS66044 (*pif5*), CS68812 (*pif3pif5*), CS66048 (*pif3pif4pif5*) and CS68096 (*pif4pif5*)) were used in the present study. The plant seeds were obtained from the Ohio State University Stock Center (Columbus, Ohio, USA). The plants were germinated for 7 days at 24 ± 1 °C during the day and 21 ± 1 °C at night (8 h photoperiod) under white fluorescent lamps (Philips, Pila, Poland) (130 μmol (photons) m^−2^ s^−1^ LI-COR LI-250A light meter (Lincoln, NE, USA). The plants were grown in 5 × 5 × 5 cm vessels filled with perlite. Throughout the growing season, the plants were watered with a two-fold diluted Hoagland nutrient solution. Then, the plants were grown for three weeks under white (WL, 450 nm + 660 nm, colour temperature 3000 K), blue (BL, 450 nm) and red (RL, 660 nm) LEDs (50 W LEDs (Epistar, Taiwan) at 24 ± 1 °C during the day and 21 ± 1 °C at night (8 h photoperiod). The emission spectra of all light sources are shown (Figure 1).

### 2.2. Measurements of CO_2_ Gas Exchange

The photosynthetic rate (P_max_) was determined in a closed system under light conditions using an LCPro + portable infrared gas analyser from ADC BioScientific Ltd. (Geddings Road, Hoddesdon, United Kingdom) connected to a leaf chamber. The CO_2_ uptake per leaf area (μmol m^−2^ s^−1^) was determined. The rate of photosynthesis of the leaves in the second layer from the top was determined at a saturating light intensity of 600 μmol (photons) m^−2^ s^−1^. For the photosynthetic measurements, fully developed, healthy-looking upper leaves with almost horizontal leaf blades were used. Each experimental variant used five to six healthy developed upper leaves from 3 or 4 plants.

### 2.3. Determination of Photochemical Activity

Fluorescence parameters characterizing the state of the photosynthetic apparatus were calculated on the basis of induction fluorescence curves plotted using data from the JIP test, which is usually used to evaluate the state of PSII. Chlorophyll (Chl) fluorescence induction curves (OJIP curves) were recorded with the Plant Efficiency Analyser (Handy-PEA, Hansatech Instruments Ltd., Narborough Road, United Kingdom). For the JIP test, OJIP curves were measured under illumination with blue light at an intensity of 6000 μmol (photons) m^−2^ s^−1^ for 1 s.

On the basis of induction fluorescence curves (OJIP curves), the following parameters, which characterize PSII photochemical activity, were calculated: F_v_/F_m_, the PSII maximum quantum photochemical yield, and PI_ABS_, the PSII performance index [21,22]. Here, F_v_ is the value of variable fluorescence, equal to the difference between F_m_ and F_0_. F_0_ is the minimum amplitude of fluorescence (F), and F_m_ is the maximum amplitude of fluorescence. For calculation of the PI_ABS_, the following formula was used: PI_ABS_ = (F_v_/F_m_)/(M_0_/V_j_) × (F_v_/F_0_) × (1 − V_j_)/V_j_); M_0_ = 4 × (F_300µs_ − F_0_)/(F_m_ − F_0_); and V_j_ = (F_2ms_ − F_0_)/(F_m_ − F_0_), where M_0_ is the average value of the initial slope of the relative variable fluorescence of Chl *a*, which reflects the closing rate of the PSII reaction centres, and V_j_ is the relative level of fluorescence in phase J after 2 ms.

PAM fluorometry (Junior-PAM, Walz, Eichenring, Germany) was used to evaluate the photosynthetic apparatus state. The values of F_0_, F_v_, F_m_, Fm′, and F′, as well as the PSII maximum (F_v_/F_m_) and effective Y(II) (F_m_′ − F_t_)/F_m_′ photochemical quantum yields and nonphotochemical quenching (NPQ) (F_m_/F_m_′ − 1), were determined. Here, F_m_ and F_m_ ′ are the maximum Chl fluorescence levels under dark- and light-adapted conditions, respectively. F_v_ is the photoinduced change in fluorescence, and F_t_ is the level of fluorescence before a saturation impulse is applied. F_0_ is the initial Chl fluorescence level. Actinic light was switched on for 10 min (I = 125 μmol (photons) m^−2^ s^−1^). For the fluorescent measurements, fully developed, healthy-looking upper leaves with almost horizontal leaf blades were used. Each experimental variant used five to six healthy developed upper leaves from 3 or 4 plants. All experiments were repeated four times (*n* = 3).

### 2.4. RNA and MicroRNA Extraction and qRT-PCR

RNA isolation was performed according to the TRIzol method. The quantity and quality of the total RNA were determined using a NanoDrop 2000 spectrophotometer (Thermo Fisher Scientific, Waltham, Massachusetts, USA). cDNA synthesis was performed using the M-MLV Reverse Transcriptase Kit (Fermentas, Waltham, Massachusetts, USA) and the oligo (dT) 21 primer nuclear coding genes. The expression patterns of the genes were assessed using the CFX96 Touch™ Real-Time PCR Detection System (Bio-Rad, Alfred Nobel Drive, Hercules, California, USA).

MicroRNA extraction was performed with a mirPremier microRNA isolation kit (Sigma, St. Louis, MO, USA). The expression patterns of microRNAs were assessed with real-time PCR using a QuantStudio 1 real-time PCR system (Thermo Fisher Scientific, Waltham, MA, USA). cDNA was synthesized with ligation-mediated reverse transcription using a miScript Plant RT Kit (Qiagen, Venlo, Limburg, The Netherlands) according to the manufacturer’s protocol. qRT-PCRs were performed according to the manufacturer’s instructions (miScript SYBR Green PCR Kit, Qiagen, Venlo, Limburg, The Netherlands) using miScript universal primer and miRNA specific primer (Appendix A) and cDNA template.

Gene-specific primers were selected using nucleotide sequences from the NCBI database (National Center for Biotechnology Information, www.ncbi.nlm.nih.gov (accessed on 20 January 2020), Bethesda, MD, USA) and the databases www.uniprot.org (accessed on 20 January 2020) and http://www.mirbase.org/database (accessed on 20 January 2020) (Manchester, UK) with Vector NTI Suite 9 software (Invitrogen, Carlsbad, CA, USA). The gene expression patterns of the protein-coding genes were normalized to the expression of the *Actin2* gene, and the microRNA expression levels were normalized to the expression of small nucleolar RNA U6.

### 2.5. Protein Extraction and Western Blotting

Total protein extracts were obtained from rosette leaves of 4-week-old plants. One hundred milligrams of tissue were collected and ground in 500 μL of extraction buffer (50 mM Tris-HCl pH 6.8, 150 mM NaCl, 0.5% Triton X-100, 1 mM DTT, 0.7% LDS, 50 μM MG132, 3 mM PMSF) and 1× protease inhibitor cocktail (F. Hoffmann-La Roche), heated for 10 min at 65 °C and cleared by centrifugation at 16,000× *g* for 20 min at 4 °C. Proteins were precipitated from supernatants with an equal volume of ice-cold 20% TCA in the presence of 0.1% β-ME for at least 1 h at −20 °C and harvested using centrifugation. The pellets were washed twice with ice-cold 80% acetone, air-dried and resuspended in 100 μL of 40 mM Tris HCl pH 6.8, 8 M urea, 0.1 mM EDTA, 1% LDS, 80 μM MG132, and 1× protease inhibitor cocktail.

For immunoblot analyses, 50 μg of denatured total proteins were separated on 8% Bis-Tris SDS-PAGE, blotted onto polyvinylidene difluoride (PVDF) membranes using semidry (Bio-Rad, Alfred Nobel Drive, Hercules, CA, USA) and probed with anti-PIF3 (AS163954), PIF4 (AS163955) and PIF5 (AS122112) antibodies according to the manufacturer’s instructions (Agrisera, Vännäs, Sweden). For the secondary antibody, horseradish peroxidase-conjugated anti-rabbit IgG (1:25,000; AS09602) for anti-PIF5 or HRP-conjugated anti-goat IgG (1:10,000; AS101068) for anti-PIF3 and anti-PIF4 were used. Blots were developed with an Agrisera ECL kit (Bright/SuperBright) (AS16 ECL-SN; Agrisera, Vännäs, Sweden) and scanned using Invitrogen iBright Imaging Systems (Thermo Fisher Scientific, Waltham, MA, USA).

### 2.6. Statistics

The experiments were performed in three biological replicates and three analytical replicates. The expression level of each gene was measured in three independent experiments. For each of these experiments, at least three parallel independent measurements were performed unless otherwise stated. The data were statistically analysed using SigmaPlot 12.3 (Systat Software, San Jose, CA, USA) with one-way analysis of variance (ANOVA) followed by Duncan’s method for normally distributed data. Different letters indicate significance at *p* < 0.05. In the analysis of microRNA expression as well as protein encoding genes, a change of two or more times was considered significant (log_2_ was greater or less than 1). The data are shown as the mean ± SD (*n* = 3).

## 3. Results

### 3.1. Morphology

The spectral composition of light had a significant effect on the size and morphology of the leaf blades of the *pif* mutants used in the experiment. Under the WL spectrum, the leaves of WT plants had a serrated edge, and *pif4* had a smoother leaf edge, while the *pif3pif5* and *pif3pif4pif5* mutants had a smooth leaf edge. The leaf petiole was elongated under WL, except in the *pif4pif5* mutant. Under BL, WT had a greater smoothness of the edge of the leaf blade; however, a similar effect was observed only in the *pif3pif5* mutant, while in all other *pif* mutants, the serrated edge of the leaves, on the contrary, was more noticeable. In *pif5* and *pif3pif5,* a decrease in leaf area was observed (Table 1).

Under the RL spectrum WT, *pif4* and *pif4pif5*, the leaf blade expanded along the entire length of the petiole, the serrated edge of the leaf blade disappeared, and leaf pubescence increased (Figure 2). At the same time, in *pif5*, *pif3pif5* and *pif3pif4pif5*, the expansion of the leaf blade along the entire petiole was expressed to a much lesser extent (Figure 2). In *pif5* under the RL spectrum, the leaf area was reduced relative to other mutants on RL (Table 1, Figure 2).

### 3.2. Gene Expression

#### 3.2.1. PIF Gene Expression

*PIF3* expression increased two-fold in the *pif4pif5* mutant under WL and RL; in all other mutants and light spectra, it was significantly reduced. The level of *PIF4* transcripts increased four-fold in the WT under WL and RL, while in all other studied mutants and light spectra, *PIF4* expression was reduced (Figure 3). The expression of the *PIF5* gene increased in the WT by more than two-fold under all light spectra. Under the BL spectrum, *PIF5* expression increased more than four-fold in the *pif4* mutant. In all other mutants, *PIF5* expression was significantly reduced. The expression of the *PIF7* gene increased two-fold under WL in the *pif4*. Under the BL spectrum, the maximum expression of *PIF7* was observed in the *pif4* mutant (four-fold increase) as well as in the WT (two-fold increase) (Figure 3). In all other mutants and light spectra, a significant decrease in *PIF7* expression was observed.

#### 3.2.2. Light Signalling TFs Expression

Under BL, *SHW1* expression increased four-fold in the *pif4* mutant and two-fold in WT, while *pif5* expression of this TF was absent. RL caused a two-fold increase in *SHW1* expression only in WT. In all other mutants and light spectra, a decrease in the levels of *SHW1* gene transcripts was observed. An increase in the expression level of the *GBF1* gene under BL was observed only in the *pif4* mutant (three-fold), against the background of a two-fold decrease in the expression of this gene in the *pif5* mutant, while in all other mutants and light spectra, a decrease in expression was observed (Figure 3). *FHY1* expression increased by two or more times in the WT under WL. Under BL, *FHY1* expression increased more than four-fold in the *pif4* mutant, while there were no changes in *FHY1* expression in the *pif5* mutant. Under RL, *FHY1* expression increased only in the WT by more than four times, while in all other mutants and light spectra, expression decreased. The transcript level of *FHY3* under WL increased almost four-fold in the WT (Figure 3). Under BL, *FHY3* expression increased more than two-fold in the WT and almost six-fold in the *pif4* mutant. In addition, in the *pif5* mutant, *FHY3* expression did not change. Under RL, *FHY3* expression increased only in the WT (more than four-fold), while in all other mutants and light spectra, expression was reduced. The expression level of *AGO1* increased under WL by two or more times in *pif5* (Figure 3).

#### 3.2.3. MicroRNA Processing Gene Expression

It is also worth noting the increase in AGO1 expression in *pif4* against the background of unchanged *pif5* expression under BL. In all other mutants and light spectra, *AGO1* expression was significantly reduced (Figure 3). The levels of *HYL1* transcripts under WL increased in the WT and *pif5* by almost two-fold. Under BL, *HYL1* expression remained at a high level in WT, while it decreased in all other mutants. The transcript level of *SPL7* increased under BL in the *pif4* mutant but did not change in *pif5* in any other mutants and light spectra expression and either did not change significantly or decreased. Under BL, *HEN1* expression increased two-fold in WT, as well as in the *pif4* mutant, while in *pif5* it remained unchanged. RL did not cause significant changes in *HEN1* expression (Figure 3).

### 3.3. miRNA Expression

During these experiments, we analysed the expression levels of some light-dependent miRNAs. First, significant increases in most of the studied miRNAs in the *pif4* mutant under BL, as well as in the *pif5* mutant under RL, should be noted.

Under the WL spectrum, the highest expression of miR156, miR166, miR167, miR168, miR170, miR396, miR404, miR472, miR833 and miR858 was observed in WT compared to other mutants. Thus, the expression of miR168, miR833 and miR858 in WT under WL was more than 30 times higher than that in other PIF mutants (Figure 4). Under the WL spectrum, WT and all mutants, except for *pif4pif5* miR408 expression, was reduced. Under the WL spectrum in *pif5* and *pif3pif4pif5* mutants, the expression of miR163 was decreased more than 10 times relative to other mutants and WT (Figure 4).

Under the BL spectrum, microRNA expression was maximal in *pif4* mutants. Among them, the highest expression of miR156, miR166, miR168, miR170, miR172, miR396, miR402, miR472, miR833 and miR858 in *pif4* under BL exceeded the other mutants by 30 or more times (Figure 4). At the same time, the expression of miR156, miR168, miR833 and miR858 in WT under BL was also high but was slightly lost in the *pif4* mutant (Figure 4).

Under the RL spectrum, the expression of the studied microRNAs was slightly decreased in the pif4 mutant, relative to the BL condition. At the same time, the expression of miR156, miR166, miR168, miR172, miR402, miR472, miR827 and miR858 in the pif5 mutant under RL increased more than 10-fold relative to the pif5 mutant under the BL spectrum. In the pif4 mutant, miR156, miR170 and miR 858 remained at a high level of expression relative to other mutants, while a decrease in the expression was observed compared to pif4 under the BL spectrum. In WT under RL, miR156, miR168, miR396 and miR858 remained at a high level relative to other mutants and light spectrum (Figure 4). It is worth paying attention to the expression levels of miR827 and miR833, because under BL, the expression level of miR827 was significantly reduced, and in pif5 under RL, on the contrary, there was an increase of more than 10 times relative to other mutants and light spectra (Figure 4). At the same time, miR833 expression increased by more than 30 times in pif4 under BL, while under RL in pif5, the decrease relative to other light variants and mutants was maximum (Figure 4).

### 3.4. Western Blot of the Proteins PIF3, PIF4, and PIF5

During the experiments, the TF PIF3, PIF4 and PIF5 protein contents in the total protein extracts of various *pif* mutants were analysed. In the WT, PIF3 and PIF5 proteins were present in all studied light variants, while PIF4 appeared only under the RL spectrum (Figure 5). In the *pif4* mutant, TF PIF4 proteins were not detected in all light variants. Additionally, in the *pif4* mutant, the TF PIF3 and PIF5 proteins were found under the BL spectrum, which was similar to the control; at the same time, under the WL spectrum, this mutant had a reduced content of the PIF3 protein. In the *pif5* mutant, the PIF5 protein was detected, but its content was reduced relative to the WT control. In contrast, PIF4 was found in all studied light spectra, which was a distinctive feature of the *pif5* mutant (Figure 5). In the double mutant *pif4pif5*, the content of all studied TF proteins was reduced, and PIF4 was absent under all light spectra. In the double mutant *pif3pif5* under WL and RL spectra, the highest content of PIF4 TF was observed, while under the RL spectrum, the protein content was reduced (Figure 5).

### 3.5. Chlorophyll Fluorescence, Photosynthetic Rate, and Leaf Area

The rate of photosynthesis (P_max_) under WL in the WT, *pif4* and *pif3pif5* mutants was lower (4–7 μmol CO_2_ m^−2^ s^−1^) than that in the *pif5*, *pif4pif5* and *pif3pif4pif5* mutants (10–11 μmol CO_2_ m^−2^ s^−1^). Under BL, the intensity of photosynthesis was the lowest in WT, *pif4* and *pif5* (4.4–6.9 μmol CO_2_ m^−2^ s^−1^), while in all other mutants, the differences in photosynthetic rate were insignificant and amounted to 8.1–8.6 μmol CO_2_ m^−2^ s^−1^. Under RL, the highest intensity of photosynthesis was in the *pif3pif4pif5* mutant (12.6 μmol CO_2_ m^−2^ s^−1^), and in other mutants, these values were lower (7–9 μmol CO_2_ m^−2^ s^−1^) (Table 1).

The PI_ABC_ indicator of the efficiency of PSII functioning under WL was the highest in the *pif3pif4pif5* mutant (3.9) and the lowest in *pif5* (2.7). The BL and RL values of PI_ABC_ did not differ markedly between the options (4.7–5.2). The effective quantum yield of PSII (Y(II)) under WL was lower in *pif5* (0.40) than in WT, while under BL, the value of Y(II) was maximal in the *pif4* and *pif3pif4pif5* mutants and minimal in the *pif5* mutants (0.44–0.51). Under RL, only *pif4*, *pif4pif5* and the WT Y(II) were reduced (0.41–0.44), and the highest Y(II) index was in *pif5* (0.52). The value of nonphotochemical NPQ quenching in *pif4pif5* plants grown under the WL spectrum was almost two times higher than that in the WT plants. Under BL, the NPQ index was maximal in the *pif5* mutant and minimal in the WT and *pif3pif5* mutant. In plants grown under RL, the value of NPQ was maximal in the *pif3pif4pif5* mutant and minimal in the *pif4* and *pif5* mutants (Table 1).

The WT had the largest leaf area regardless of the light spectrum. RL caused an increase in the leaf area in *pif4*, *pif4pif5* and *pif3pif4pif5*, but not in *pif5*, where the leaf area was significantly reduced (5.1 ± 3.2 cm^2^). Under BL, the leaf area was increased in the *pif4pif5* and *pif3pif4pif5* mutants, while in *pif4*, *pif5* and *pif3pif5* it was reduced relative to other mutants. WL influenced the increase in the leaf area in the *pif5* mutant, and in the remaining mutants it was reduced relative to WT and *pif5* (Table 1).

## 4. Discussion

### 4.1. Photosynthesis and Morphology

PIF TFs are involved in light signalling [23,24], and, together with phytochromes, they influence the numerous reactions of plants to light spectra [25]. For example, PIF4 plays a negative role in RL-mediated PHYB signalling [5], and PIFs are involved in the activation of cell elongation [26]. PIFs interact with light-activated photoreceptors via a conserved phyB-binding domain [25,27,28]. PIF4 affects light signalling, which can both affect plant photomorphogenesis and change the rate of photosynthesis. In our experiments, in the *pif5*, *pif4pif5* and *pif3pif4pif5* mutants of *A. thaliana* under WL and in the *pif3pif4pif5* mutant under RL, an increase in the photosynthetic rate relative to the WT control was observed (Table 1). In contrast, PSII activity, evaluated by PI_ABC_ and Y(II) was activated by BL, whereas the values of PI_ABC_ and Y(II) did not differ between mutants. It is known that the difference between the electron transport rate (ETR) or Y(II) and the net CO_2_ assimilation rate (P_max_) is an indicator of the existence of alternative electron sinks or, as an example, nutrition limitations as well as water shortages [22,29]. We found different effects of PIFs on the independent responses of photosynthesis and morphology to light spectra and did not observe a correlation between fluorescence parameters and P_max_, which is likely linked to the existence of alternatives to CO_2_ assimilation electron sinks. Parameters such as Y(II) and ETR vary greatly depending on the light and other conditions, but the value of F_v_/F_m_ is more conservative [22]. In addition, the measurements of P_max_ were provided under light saturating conditions and Y(II) and ETR—under growth light intensity. The absence of a correlation between parameters such as F_v_/F_m_ and Y(II) is likely linked to different light spectra and PIFs deficiency when the correlation can be destroyed. This is probably because BL inactivates phytochrome signalling, and the differences between the mutants are smoothed out, while the differences are more noticeable under WL and RL, where phytochromes are more active (Table 1). Comparison of photosynthetic activity and the phenotype of *A. thaliana pif* mutants indicates a loss of connection between photosynthesis and phenotype. Thus, under BL spectrum, in the mutants *pif4* and *pif5*, as well as *pif3pif5*, the leaf area decreased by more than two times relative to the WT, while the intensity of photosynthesis remained stable. The changes in phenotype are mainly linked to the action of PIF deficit on photomorphogenesis.

### 4.2. Main Differences in MicroRNA and TFs Expression

PIFs are able to regulate the expression of MIR genes and, as a result, change plant photomorphogenesis. Thus, MIR156 is the target gene for PIF5 [23,28]. PIF5 suppresses the transcription of MIR156 by directly binding to the promoter sites of the genes of this miRNA. This causes a decrease in mature miR156 levels and a coordinated increase in miR156 target SPL7 transcripts. In our experiments on adult plants under BL, the increase in miR expression was more important, not so much for the PIF5 protein content, but rather for the absence of the PIF4 protein (Figure 5). At the same time, SPL7 expression and miR156 expression were maximal in the *pif4* mutant under BL (Figure 3 and Figure 4). In *pif* mutants, a violation of the phyB-Pfr-PIF interaction is assumed, which should be accompanied by a decrease in the regulatory ability of PIFs. Previously, Sun and co-authors showed that when plants are transferred from darkness to RL, the expression of most light-dependent microRNAs in *A. thaliana* seedlings changes in the *pif4* mutant. PIF4 integrates RL signalling and miRNA biogenesis through regulation of transcription and miRNA processing, influencing DCL1 stability. At the same time, PIF4 can directly bind to the promoters of a group of microRNA genes and control their transcription in seedlings [4]. In our studies, the highest activation of almost all studied miRNAs was observed in the *pif4* mutant under BL, accompanied by an increase in the content of the PIF5 protein (Figure 4). In some cases, the increase in expression in *pif4* under BL was 10–30 times higher than in other mutants and light treatments (Figure 4). An increase in the expression of most conserved miRNAs (miR156, miR160, miR163, miR165, miR166, miR167, miR168, miR170 and miR172) correlated with elongation of the leaf petiole, as well as the appearance of a serrated edge of the leaf blade (Figure 2). Auxins, as well as auxin signalling factors, are involved in the formation of serrated leaves, which may be regulated by miRNAs under the BL spectrum [13,30,31]. Another feature of the *pif4* mutant under the BL spectrum was an increase in miR408 against the background of a decrease in miR827. In other works, it was reported that an increase in miR408 leads to an increase in the productivity and photosynthesis of *A. thaliana* plants [32], which was not observed in our experiments. Mir827 is involved in the homeostasis of mineral nutrition elements and cell aging processes [33]. An increase in miRNA expression was observed in *pif5* under RL, but it was less pronounced than in *pif4* under BL (Figure 4). Changes in miRNA expression in the *pif5* mutant in response to RL manifested as changes in leaf shape as well as leaf size (Figure 2 and Figure 4; Table 1). As a result, we were able to detect microRNAs whose expression increased only in *pif4* under the BL spectrum including miR163, miR319, miR398, miR408 and miR833, while miR827 decreased. We suggest that there is an inverse relationship between these microRNAs and the PIF4 protein. This assumption is additionally confirmed by the fact that under the RL spectrum in *pif5* mutants, practically the same miRNAs decrease the expression of miR163, miR319, miR398, miR408 and miR833, while mi827, on the contrary, increased, against the background of an increase in the amount of PIF4 protein in the *pif5* mutant (Figure 3 and Figure 5). It is difficult to say exactly to what extent PIF3 is involved in the regulation of miRNAs, because in our experiments, PIF3 protein was found in the samples of *pif3* mutants, which indicates the possibility of synthesizing this protein at later stages of ontogeny even in homozygous mutants. At the same time, miR167, miR168, miR396, miR472, miR858 and miR402 increased in both BL and RL, although under RL, the increase was less noticeable (Figure 4). These miRNAs, in our opinion, do not have light-dependent regulation by PIF in adult *A. thaliana* plants.

We found that the expression of some miRNAs is upregulated (miR163, miR319, miR398, miR408, miR833), while others are downregulated (miR827), indicating that PIF4 can serve as both a positive and negative transcriptional regulator for different miRNA genes in both seedlings and adult plants. A similar effect was observed in the *pif4pif5* double mutant, which can be explained by the different abilities of PIF4 and PIF5 to change MIR gene transcription, as well as the ability of other PIFs or TF to influence microRNA expression (Figure 3 and Figure 4). This interpretation is also supported by the fact that the *pif4* mutant shows activation of transcription of other TFs *PIF7*, *SHW1*, *GBF1*, *FHY1* and *FHY3*, which was not observed under BL in other mutants and the WT (Figure 3). Some of these TFs are negative regulators and may indirectly influence the formation of mature microRNAs. In addition, the expression of most of the studied miRNAs in the *pif4* and *pif5* mutants also increased under RL, but increases in the TF transcripts *PIF7*, *SHW1*, *GBF1*, *FHY1* and *FHY3* were not observed (Figure 3).

### 4.3. MicroRNA Processing Genes

As mentioned above, microRNA biogenesis and processing proteins such as AGO, DCL1, HYL1 and HEN1 are also involved in miRNA-mediated photomorphogenesis [34]. *A. thaliana ago1* mutants are hypersensitive to light, probably due to dysregulation of the PHYA-dependent light signalling pathway [7]. Moreover, the de-etiolation of *A. thaliana ago1-27* mutants under the FR spectrum is impaired, which indicates that AGO1 is required for normal photomorphogenesis, at least in the early stages of ontogeny [35]. Since AGO1 is a key component of RISC, it can be assumed that miRNA is involved in responses to FR. In our experiments, the expression of *AGO1*, *HYL1* and *HEN1* increased only in the *pif4* mutant under BL, accompanied by an increase in the amount of mature miRNAs (Figure 3 and Figure 4). This finding can be explained by the fact that miRNAs compete for binding to AGO proteins, and intense miRNA expression can affect AGO1 expression. In addition, miR160 downregulated AGO1 transcripts, an expression that was upregulated in *pif4* and in BL and RL (Figure 3). The proteins involved in miRNA processing, DCL1 and HYL1, are also able to interact with light-dependent TFs of the bHLH (basic helix–loop–helix) family (to which PIF4 belongs) and destabilize DCL1 during plant irradiation with the RL spectrum [4], which was also confirmed in our studies. Moreover, the greatest activation of these processes occurred under BL (Figure 3). In addition, it has been shown that under the BL spectrum, another photoreceptor, CRY1, is able to directly interact with PIF4 and suppress its interaction with PHYB, suggesting the presence of coordination between blue (CRY1) and red light (PHYB) signals [19].

## 5. Conclusions

The highest activation of the majority of studied miRNAs was observed in the *pif4* mutant under BL, accompanied by an increase in the content of the PIF5 protein. A less significant increase in microRNA expression was observed in the *pif5* mutant under RL. We assume that the increase in the PIF5 content in the absence of PIF4 ensures the activation of light-dependent miRNAs. Activation of the formation of mature miRNAs under BL in the *pif4* mutant is also associated with an increase in the transcription of light-dependent TFs. We found that the expression of BL-light-dependent microRNAs (miR163, miR319, miR398, miR408, miR833) increased only in the *pif4* mutant, while miR167, miR168, miR396, miR397, miR402, miR472 and miR858 were less regulated by PIF4 and PIF5 under BL and RL. We observed an increase in the expression of most of the studied microRNAs in the *pif4* mutant under the BL spectrum accompanied by morphological changes, but we did not observe an increase in the photosynthetic rate compared to the RL and WL spectra. It is likely that not only a decrease in the activity of the phytochrome system and all associated regulatory and signalling molecules under these spectra, but also the activation of cryptochromes and/or phototropins, affects the processing of mature miRNAs. In this case, PIF4 and PIF5 are of the greatest importance, providing possible cross-regulation between light receptors of different spectra. These studies have revealed previously unknown interactions between miRNA biogenesis and light signalling under different spectral compositions through PIF-dependent regulation at BL in adult *A. thaliana* plants.

## Figures and Tables

**Figure 1 cells-11-03981-f001:**
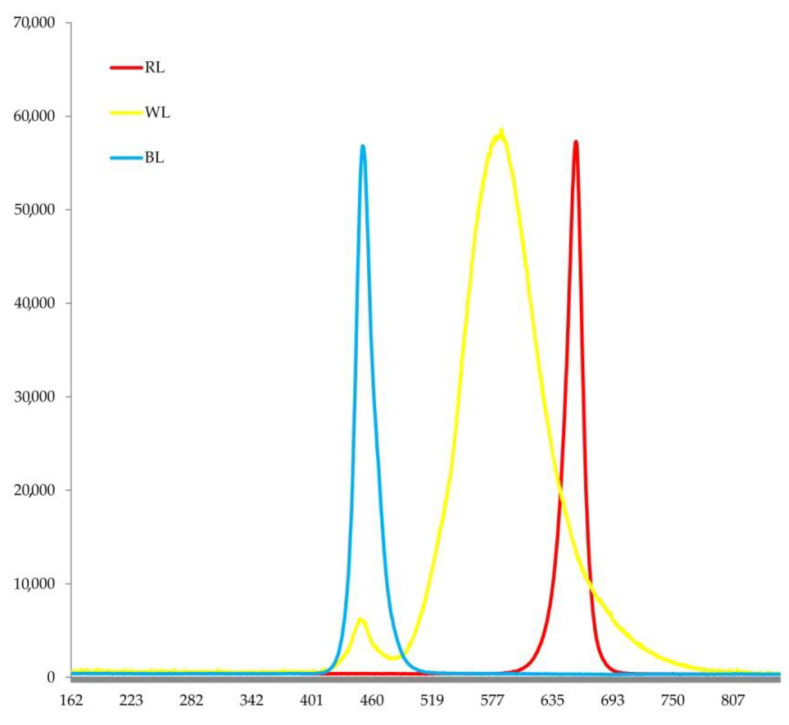
Emission spectra of various light sources used in the experiments, white light (WL), blue light (BL) and red light (RL).

**Figure 2 cells-11-03981-f002:**
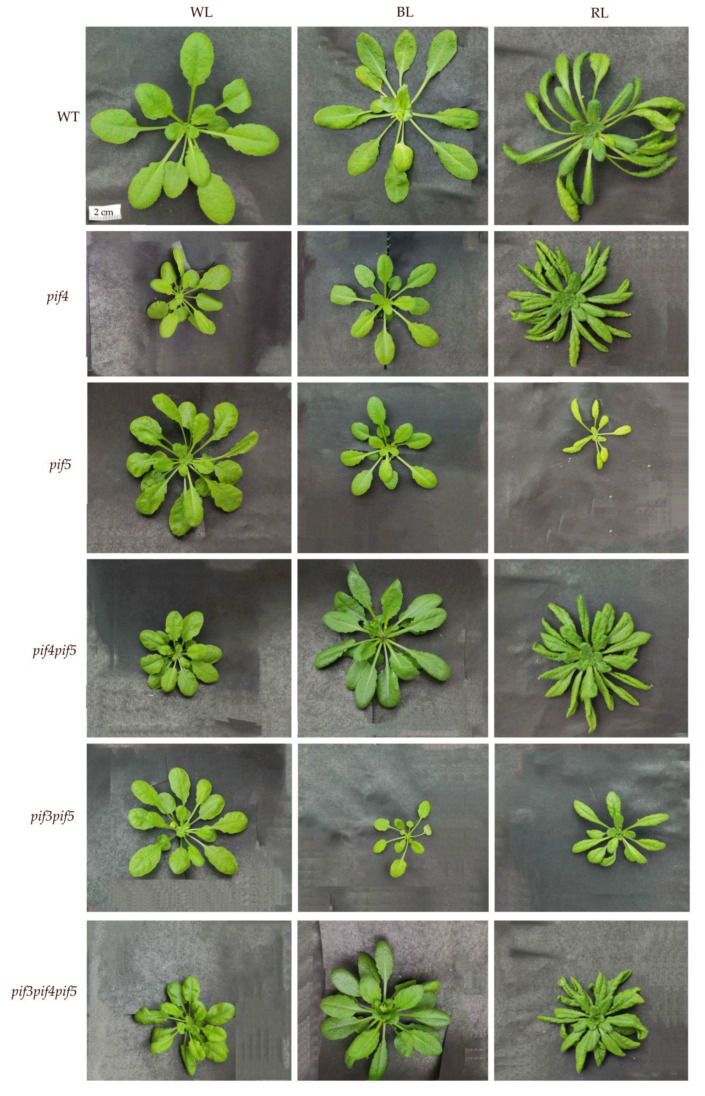
Effect of light with different spectra on the phenotype of four-week-old *A. thaliana pif* mutants.

**Figure 3 cells-11-03981-f003:**
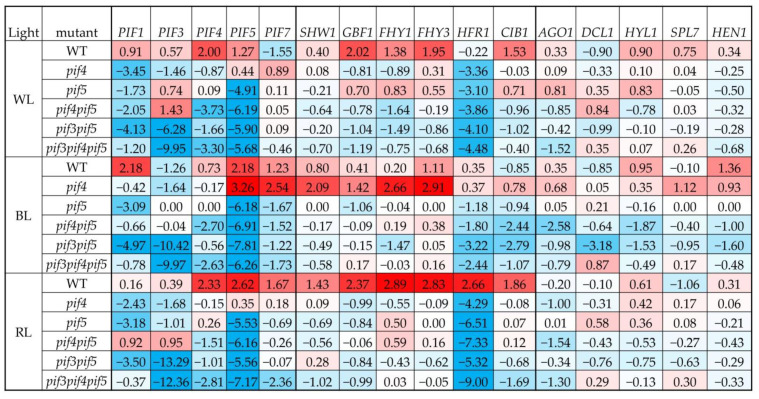
Effect of light with different spectra on the expression of transcription factor genes, light signalling genes, and microRNA processing genes in wild-type and *pif* mutant four-week-old *A. thaliana* plants. Log_2_ data are presented. Changes were considered significant if the expression was above or below one (*p* < 0.05; *n* = 3). The transcript levels were normalized to the expression of the *Actin1* gene.

**Figure 4 cells-11-03981-f004:**
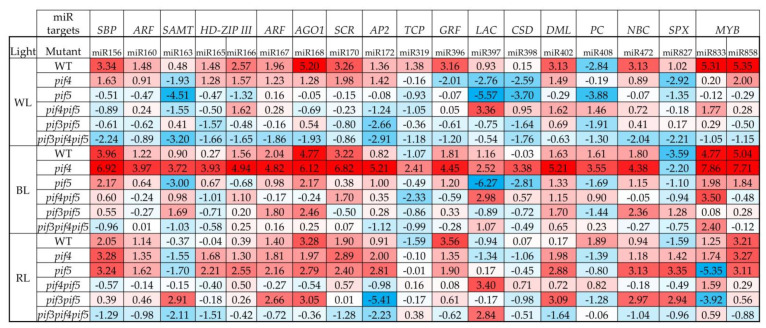
Effect of light with different spectra on the expression of light-dependent miRNAs in the WT and *pif* mutant 4-week-old *A. thaliana* plants. Log_2_ data are presented. Changes were considered significant if the expression was above or below one (*p* < 0.05; *n* = 3). MicroRNAs are marked with a black marker, which, according to the results of our studies, is dependent on PIF4 under the BL spectrum, and the expression of which changes in the *pif5* mutant under the RL spectrum where the PIF4 protein is present. The top line of Figure 4 shows the target genes of the studied miRNAs (squamosa promoter binding proteins (*SBP*), auxin response factors (*ARF*), S-adenosyl-Met-dependent carboxyl methyltransferase (*SAMT*), HD-ZIP III transcription factor (*HD-ZIP III*), Argonaute protein 1 (*AGO1*), GRAS domain or SCARECROW-like proteins (*SCR*), APETALA2 (*AP2*), teosinte branched1/cycloidea/proliferating cell factor (*TCP*), growth-regulating factor (*GRF*), laccase (*LAC*), Cu/Zn superoxide dismutase (*CSD*), demeter-like protein (*DML*), plastocyanin (*PC*), CC-NBS-LRR gene family (*NBC*), SPX -domain-containing genes, nitrogen limitation adaptation (NLA) and phosphate transporter 5 (PHT5) (*SPX*), MYB transcription factors (*MYB*)).

**Figure 5 cells-11-03981-f005:**
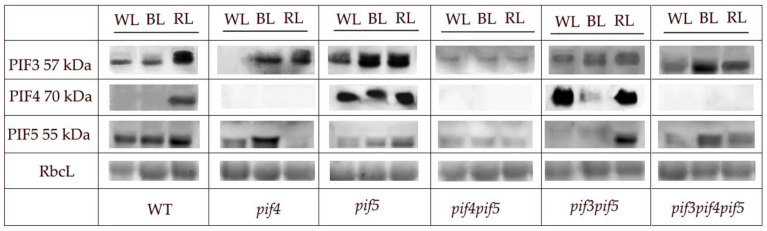
Immunoblot analyses of proteins extracted from rosette leaves of *A. thaliana pif* mutants grown under light with different spectra (anti-PIF3 (AS163954), anti-PIF4 (AS163955) and anti-PIF5 (AS122112) antibodies. The Ponceau S-coloured level of RbcL protein is shown as a loading control.

**Table 1 cells-11-03981-t001:** Influence of WL, BL and RL on the main indicators fluorescence chlorophyll DI0/RC dissipated energy flux per reaction centre, Y(II)—PSII effective quantum yield; PI_abc_—PSII performance index; NPQ, nonphotochemical fluorescence; F_v_/F_m_—PSII maximum quantum yield and photosynthetic rate P_max_ (µmol CO_2_ m^−2^ s^−1^), as well as leaf area (cm^2^) in leaves above-ground part of *A. thaliana pif* mutants. Values are the mean ± SE. Different letters denote statistically significant differences in the means at *p* < 0.05 (ANOVA followed by Duncan’s method).

WL	WT	*pif4*	*pif5*	*pif4pif5*	*pif3pif5*	*pif3pif4pif5*
DI0/RC	0.53 ± 0.03 ^b^	0.56 ± 0.02 ^b^	0.76 ± 0.05 ^a^	0.59 ± 0.03 ^b^	0.59 ± 0.02 ^b^	0.51 ± 0.03 ^b^
PIabs	3.35 ± 0.32 ^ab^	3.35 ± 0.32 ^ab^	2.66 ± 0.14 ^b^	3.09 ± 0.21 ^ab^	3.23 ± 0.20 ^ab^	3.91 ± 0.31 ^a^
Y(II)	0.47 ± 0.02 ^a^	0.42 ± 0.02 ^ab^	0.40 ± 0.01 ^b^	0.45 ± 0.01 ^ab^	0.42 ± 0.03 ^ab^	0.43 ± 0.01 ^ab^
NPQ	0.69 ± 0.04 ^c^	0.97 ± 0.02 ^ab^	0.89 ± 0.03 ^b^	0.91 ± 0.01 ^b^	1.05 ± 0.02 ^a^	0.65 ± 0.02 ^c^
F_v_/F_m_	0.81 ± 0.03 ^a^	0.78 ± 0.02 ^a^	0.79 ± 0.02 ^a^	0.80 ± 0.01 ^a^	0.80 ± 0.01 ^a^	0.81 ± 0.02 ^a^
P_max_, µmol CO_2_ m^−2^ s^−1^	6.63 ± 0.54 ^b^	5.73 ± 0.09 ^b^	10.5 ± 0.24 ^a^	10.16 ± 1.41 ^a^	4.36 ± 0.32 ^c^	10.26 ± 0.75 ^a^
Leaf area, cm^2^	35.1 ± 2.3 ^a^	13.7 ± 2.8 ^c^	33.7 ± 1.4 ^a^	17.5 ± 3.2 ^c^	25.1 ± 1.9 ^b^	20.5 ± 2.7 ^bc^
BL	WT	*pif4*	*pif5*	*pif4pif5*	*pif3pif5*	*pif3pif4pif5*
DI0/RC	0.44 ± 0.03 ^a^	0.43 ± 0.01 ^a^	0.47 ± 0.03 ^a^	0.43 ± 0.03 ^a^	0.49 ± 0.03 ^a^	0.47 ± 0.01 ^a^
PI_abs_	4.92 ± 0.39 ^a^	4.97 ± 0.29 ^a^	4.56 ± 0.43 ^a^	4.65 ± 0.34 ^a^	4.71 ± 0.18 ^a^	5.20 ± 0.30 ^a^
Y(II)	0.48 ± 0.03 ^a^	0.50 ± 0.01 ^a^	0.44 ± 0.02 ^a^	0.49 ± 0.03 ^a^	0.47 ± 0.01 ^a^	0.51 ± 0.02 ^a^
NPQ	0.79 ± 0.03 ^c^	0.85 ± 0.02 ^bc^	1.13 ± 0.04 ^a^	0.91 ± 0.03 ^b^	0.74 ± 0.03 ^c^	0.84 ± 0.03 ^bc^
F_v_/F_m_	0.81 ± 0.01 ^a^	0.81 ± 0.01 ^a^	0.82 ± 0.01 ^a^	0.81 ± 0.01 ^a^	0.81 ± 0.01 ^a^	0.82 ± 0.01 ^a^
P_max_, µmol CO_2_ m^−2^ s^−1^	4.36 ± 0.44 ^c^	4.86 ± 0.20 ^c^	6.93 ± 0.53 ^b^	8.13 ± 0.23 ^a^	8.63 ± 0.20 ^a^	8.53 ± 0.33 ^a^
Leaf area, cm^2^	39.4 ± 3.3 ^a^	17.7 ± 3.4 ^b^	14.9 ± 3.8 ^b^	36.3 ± 2.1 ^a^	6.3 ± 2.2 ^c^	37.1 ± 2.8 ^a^
RL	WT	*pif4*	*pif5*	*pif4pif5*	*pif3pif5*	*pif3pif4pif5*
DI0/RC	0.46 ± 0.03 ^a^	0.46 ± 0.02 ^a^	0.48 ± 0.03 ^a^	0.52 ± 0.02 ^a^	0.51 ± 0.03 ^a^	0.49 ± 0.03 ^a^
PI_abs_	3.33 ± 0.37 ^a^	4.74 ± 0.46 ^a^	3.63 ± 0.34 ^a^	4.67 ± 0.38 ^a^	3.98 ± 0.35 ^a^	4.23 ± 0.32 ^a^
Y(II)	0.44 ± 0.01 ^b^	0.41 ± 0.04 ^b^	0.52 ± 0.01 ^a^	0.43 ± 0.01 ^b^	0.44 ± 0.03 ^ab^	0.45 ± 0.03 ^ab^
NPQ	0.74 ± 0.04 ^ab^	0.49 ± 0.02 ^c^	0.47 ± 0.04 ^c^	0.76 ± 0.02 ^ab^	0.63 ± 0.03 ^b^	0.83 ± 0.03 ^a^
F_v_/F_m_	0.80 ± 0.01 ^a^	0.81 ± 0.01 ^a^	0.79 ± 0.01 ^a^	0.81 ± 0.02 ^a^	0.8 0 ± 0.02 ^a^	0.82 ± 0.01 ^a^
P_max_ µmol CO_2_ m^−2^ s^−1^	8.03 ± 0.36 ^b^	6.6 ± 0.57 ^d^	9.06 ± 0.67 ^b^	8.3 ± 1.52 ^c^	7.73 ± 0.04 ^c^	12.66 ± 1.03 ^a^
Leaf area, cm^2^	37.2 ± 3.8 ^a^	33.1 ± 3.4 ^a^	5.1 ± 3.2 ^d^	30.9 ± 3.8 ^a^	10.9 ± 2.7 ^c^	24.5 ± 2.8 ^b^

## Data Availability

The datasets generated during and/or analysed during the current study are available from the corresponding author on reasonable request.

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
