# Peer review of "Independent Responses of Photosynthesis and Plant Morphology to Alterations of PIF Proteins and Light-Dependent MicroRNA Contents in Arabidopsis thaliana pif Mutants Grown under Lights of Different Spectral Compositions"

_cells, 2022, doi:10.3390/cells11243981_

Round 1
Reviewer 1 Report
My main comment: the coherence (title-objectives-results-conclusion) of the manuscript must be improved.
Comments on the MS cells-2012863 submitted by Pashkovskiy et al.
This manuscript presents a complex set of molecular components of the signaling network involved in the perception and transmission of light signals in plants, with a focus on the transcription factors interacting with phytochrome (PIFs). Although the data set is impressive, it is my opinion that several points need to be improved before its publication.
1- The rationale of the study needs to be clarified. The second last paragraph of the Introduction (li. 82-88) mentioned that the regulatory action of PIF4 on miRNA under blue light remained unexplored. This sentence failed to mention that PIF4-5 act as regulators of blue-light (BL) receptors PHOT1 and CRY1, increasing the likeliness of the BL effects on PIF4 and miRNA. Further in this paragraph (li. 84-88), the authors justified their study by using mature plants instead of seedlings. However, the effects of light conditions on PIFs in different mutants were not compared at these two growth stages.
2- The comments above lead to my main comment, there is a lack of coherence between the title, the objectives and the main conclusion (highlights). The title is neutral (Impact(s) of light spectra on morphology, photosynthesis, miRNAs, and PIF content), the objectives announced a relationship between miRNS, PIF gene transcripts and PIF under BL and RL, whereas the highlights affirm that PIF4 regulates miRNAs under BL. In the second sentence (li. 500-501) of the highlights, besides a poor syntax, the authors cannot affirm based on their results that light quality affects phenotype (morphology) and photosynthesis via the regulatory action of PIFs. Indeed, I was not able to see a relationship (no correlation, even less a causal relation) between PIF4 expression, morphology and photosynthesis under different lights.
In short, there is no clear “take-home message” in this manuscript. As mentioned in the Intro and in the highlights, the main contribution of this study to the PIF knowledge PIF4 regulates miRNAs under not only under RL, but even more under BL.
3- A weakness of this manuscript that could be easily improved is to better define the spectral qualities of the three light “colors” used in this study, T, not sufficiently considered in this study. To make things worse, the supplemental figure S1 was not available. To better interpret the results, it is important to know the ratios of red to far-red light intensities to estimate the proportion of active phytochromes (Pfr/Ptotal) in both white light and red light.
4- The results on plant morphology are incomplete. In line with the previous point, there is a clear effect of red light on leaf shape (more elongate), distinct from leaves developed under white light and blue light (more spatulate), irrespectively of the wt/mutants. However, there is no data on leaf area, plant biomass, ... There are large difference in plant size, but it is not clear if the pictures were taken at the same phenological stage. Two mutants under BL seem close to the flowering stage, in contrast to others.
5- The data on photosynthesis are difficult to interpret. There are no difference between the different treatments (mutants x lights) when measurements were made in darkness (DI0/RC, PIabs, Fv/Fm). There are differences between YII, NPQ, and Pn from the different treatments (lights X mutants), but no clear picture emerging. It is difficult to compare photosynthetic performances from plants grown under different light conditions when photosynthesis of these plants is measured under the same measuring light (white light? This should be indicated). Another point that obscures the picture is that there is no correlation at all (R2 = 0,0045) between YII (measuring at the growth light intensity) and Pn (measured under saturating light).
6- Globally, the text is not reader-friendly, and there are several sentences with awkward syntax.
The third sentences (li,17-20) of the abstract and of the Intro (li. 35-40) need attention, some words seem to be missing.
The word “variants” (ex. … all studied variants, li. 371) referring to the different light colors is misleading as it can be associated to the different mutants.
The first sentence of the Results section should be rewritten. Under WL, the WT and the studied mutants had a similar morphology.
The description of figures 2 and 3 is tedious, and could be better organized to guide the attention to the most important points. The first paragraph of section 3.2 ends with the PIF7 expression, and the next paragraph continue with the same idea. A line above figure 2 should be added to indicate the three different sub-sections: TF genes, light signaling genes, and miRNA genes.
The 2nd sentence of section 3.4 (li. 364-366) needs clarification.
Many sentences in the 1st paragraph of the Discussion need improvements (missing words).
Author Response
1.My main comment: the coherence (title-objectives-results-conclusion) of the manuscript must be improved.
Answer:
We are grateful to the reviewer for a thorough reading the manuscript.
We have improved the title-objectives-results-conclusion (please see text)
2. The rationale of the study needs to be clarified. The second last paragraph of the Introduction (li. 82-88) mentioned that the regulatory action of PIF4 on miRNA under blue light remained unexplored. This sentence failed to mention that PIF4-5 act as regulators of blue-light (BL) receptors PHOT1 and CRY1, increasing the likeliness of the BL effects on PIF4 and miRNA.
Answer:
We are grateful to the reviewer for pointing out this inaccuracy. We have changed the text of the manuscript.
Along with this, PIFs are also able to interact with blue light receptors, such as CRY and PHOT, but how this leads to the expression of light-dependent microRNAs is poorly understood
3. Further in this paragraph (li. 84-88), the authors justified their study by using mature plants instead of seedlings. However, the effects of light conditions on PIFs in different mutants were not compared at these two growth stages.
Answer:
We appreciate the reviewer for the comment. Indeed, in our present research we are focusing on adult plants instead of commonly used seedlings, as, for example, other research group has previously shown the mutual regulation of miRNA and PIF in Arabidopsis seedlings (https://doi.org/10.1371/journal.pgen.1007247). We have added references to the literature to make it clear what the seedlings have been explored before us. To our data, we hypothesized, since the same certain proteins may exhibit distinct functional activity at different stages of ontogenesis, perhaps, it might be similar with PIFs. We found interesting to analyze PIFs at the stage, where usually their expression level very low or undetectable under white light conditions. Therefore, we decided to focus on adult plants to elucidate potentially novel functional properties of PIF proteins, without comparing them in the seedling and adult plant stages. It should be notice that most protein extraction methods are developed for seedlings, so we had to adapt the protein extraction method for adult plants.
4. The comments above lead to my main comment, there is a lack of coherence between the title, the objectives and the main conclusion (highlights). The title is neutral (Impact(s) of light spectra on morphology, photosynthesis, miRNAs, and PIF content), the objectives announced a relationship between miRNS, PIFgene transcripts and PIF under BL and RL, whereas the highlights affirm that PIF4 regulates miRNAs under BL. In the second sentence (li. 500-501) of the highlights, besides a poor syntax, the authors cannot affirm based on their results that light quality affects phenotype (morphology) and photosynthesis via the regulatory action of PIFs. Indeed, I was not able to see a relationship (no correlation, even less a causal relation) between PIF4 expression, morphology and photosynthesis under different lights.
In short, there is no clear “take-home message” in this manuscript. As mentioned in the Intro and in the highlights, the main contribution of this study to the PIF knowledge PIF4 regulates miRNAs under not only under RL, but even more under BL.
Answer:
We are grateful to the reviewer for finding illogicalities in some parts of the manuscript.
We've fixed the highlights and conclusions.
Light quality affects on phenotype pif mutants but not photosynthetiс rate.
And we completely revised the manuscripts, tried to discuss the results more specifically
5. A weakness of this manuscript that could be easily improved is to better define the spectral qualities of the three light “colors” used in this study, T, not sufficiently considered in this study. To make things worse, the supplemental figure S1 was not available. To better interpret the results, it is important to know the ratios of red to far-red light intensities to estimate the proportion of active phytochromes (Pfr/Ptotal) in both white light and red light.
Answer: Now the figure with the spectra of the sources will be in the main text of the manuscript. In our white light sources, far-red light has been eliminated entirely.
6. The results on plant morphology are incomplete. In line with the previous point, there is a clear effect of red light on leaf shape (more elongate), distinct from leaves developed under white light and blue light (more spatulate), irrespectively of the wt/mutants. However, there is no data on leaf area, plant biomass, ... There are large difference in plant size, but it is not clear if the pictures were taken at the same phenological stage. Two mutants under BL seem close to the flowering stage, in contrast to others.
Answer:
The classic effect of blue light is accelerated flowering, however, in the analyzes we tried to use plants without peduncles or with minimal bud buds. The plants were photographed at the same age. We added a discussion of the morphological features of plants, additionally we determined the leaf areas and put these data into the table.
7. The data on photosynthesis are difficult to interpret. There are no difference between the different treatments (mutants x lights) when measurements were made in darkness (DI0/RC, PIabs, Fv/Fm). There are differences between YII, NPQ, and Pn from the different treatments (lights X mutants), but no clear picture emerging. It is difficult to compare photosynthetic performances from plants grown under different light conditions when photosynthesis of these plants is measured under the same measuring light (white light? This should be indicated). Another point that obscures the picture is that there is no correlation at all (R2 = 0,0045) between YII (measuring at the growth light intensity) and Pn (measured under saturating light).
Answer:
It is known that a difference between the electron transport rate (ETR) or Y(II) and the net CO2 assimilation rate (Pn) are an indicator of the existence of alternative electron sinks (Kalaji et al. 2014). For example, an increased ETR/Pn ratio indicates the existence of other electron sinks (e.g., Mehler reaction, photorespiration, nitrate reduction) in competition with CO2 assimilation (e.g., Kalaji et al. 2014). Also, it is known that the parameters such as Y(II) and ETR vary greatly depending on the light and other conditions but value of Fv/Fm under normal conditions is more conservative (Kalaji et al. 2014)
We added some sentences to the MS: "It is known that a difference between the electron transport rate (ETR) or Y(II) and the net CO2 assimilation rate (Pn) are an indicator of the existence of alternative electron sinks (Fryer et al. 1998;Kalaji et al. 2014)". We do not see a correlation between fluorescence parameters and Pn and It is likely linked to the existence of alternative to CO2 assimilation electron sinks. The parameters such as Y(II) and ETR vary greatly depending on the light and other conditions but value of Fv/Fm is more conservative (Kalaji et al. 2014). The absence of a correlation between the parameters such as Fv/Fm and Y(II) is likely linked to under conditions of different lights and PIFs deficiency when the correlation can be destroyed.
M.J. Fryer et al. Relationship between CO2 Assimilation, Photosynthetic Electron Transport, and Active O2 Metabolism in Leaves of Maize in the Field during Periods of Low Temperature. Plant Physiol. (1998) 116: 571–580.
Kalaji Hazem M. • Gert Schansker • Richard J. Ladle • et al. Frequently asked questions about in vivo chlorophyll fluorescence: practical issues. Photosynth Res (2014) 122:121–158. DOI 10.1007/s11120-014-0024-6.
8. Globally, the text is not reader-friendly, and there are several sentences with awkward syntax.
The third sentences (li,17-20) of the abstract and of the Intro (li. 35-40) need attention, some words seem to be missing.
Answer: We are grateful to the reviewers for their comments. We have completely redesigned the abstract
9. The word “variants” (ex. … all studied variants, li. 371) referring to the different light colors is misleading as it can be associated to the different mutants.
The first sentence of the Results section should be rewritten. Under WL, the WT and the studied mutants had a similar morphology.
Answer: We are grateful to the reviewers for their comments. We have completely redesigned the section 3.1. Morphology
10. The description of figures 2 and 3 is tedious, and could be better organized to guide the attention to the most important points.
Answer: The description of figures 2 and 3 has been completely redone according to the recommendations of the reviewer
11. The first paragraph of section 3.2 ends with the PIF7expression, and the next paragraph continue with the same idea.
Answer: Done
12. A line above figure 2 should be added to indicate the three different sub-sections: TF genes, light signaling genes, and miRNA genes.
Answer: Done
13. The 2ndsentence of section 3.4 (li. 364-366) needs clarification.
Many sentences in the 1st paragraph of the Discussion need improvements (missing words).
Answer: Done
The English text of the manuscript has been improved editing from American Journal Experts (Certificate verification key 1A0F-7257-F5BC-4C00-239F)
Reviewer 2 Report
The ms is hard to read but contains relevant and interesting data. Best regards, Ernst woltering Review cells-2012863 by Ernst woltering Authors studied the effect of light spectrum on PIF and miRNAs in arabidopsis. Three independent experiments were done. I think most things were measured at one time point. That means to me that there can be differences between treatments/mutants. This should be written as: ... was higher in .. than in ... . Or level of .. was highest in .. and lowest in .. Authors consequently talk about levels of was increased in .. and was decreased in .. But in all thhese cases it s not clear TO WHAT REFERENCE it was increased or decreased. I do not think that there has been a measurement of e.g. gene expression at say time zero and that increased refers to difference between time zero and end point. Everywhere in the ms where authors used increased or decreased they probably mean to say high or low abundance/levels. If something is "increased", it should also be stated with respect to WHAT it is increased. (WT under WL could be a reference, but that authors should clearly state that). Te description of results is very elaborate and very hard to read. Think authors should e.g. make sub paragraphs. Some more structural description would help me very much. For example - Response of WT to different light treatmenst - Response of Pif 4 to light treatmenst - Rsponse of PIF5 to light - And so on Or alternatively: - Response of WT and mutants to WL - Response of WT and mutants to Blue - Response to Red - And so on Also the discussion section is very dense. Could be helpful to make sub paragraphs here It could also be very informative if authors could make e.g. a PCA plot or something alike to visualize the gene expression and mRNA data with respect to light and with respect to mutant type. Also it would be informative to make a scheme depicting the results in a more visible way, showing based on the results what the sequence of events most probably is in response to different light colors. On the whole I think the data are interesting and it is a convincing story, but the MS is very hard to read/understand because it lacks a good structure.
Author Response
1. Three independent experiments were done. I think most things were measured at one time point. That means to me that there can be differences between treatments/mutants. This should be written as: ... was higher in .. than in ... . Or level of .. was highest in .. and lowest in .. Authors consequently talk about levels of was increased in .. and was decreased in .. But in all thhese cases it s not clear TO WHAT REFERENCE it was increased or decreased. I do not think that there has been a measurement of e.g. gene expression at say time zero and that increased refers to difference between time zero and end point. Everywhere in the ms where authors used increased or decreased they probably mean to say high or low abundance/levels. If something is "increased", it should also be stated with respect to WHAT it is increased. (WT under WL could be a reference, but that authors should clearly state that).
Answer: We are grateful to the reviewers for their careful attitude to the manuscript.
We reworked the manuscript in such a way as to satisfy the fair comment of the reviewer.
2. Te description of results is very elaborate and very hard to read. Think authors should e.g. make sub paragraphs. Some more structural description would help me very much. For example - Response of WT to different light treatmenst - Response of Pif 4 to light treatmenst - Rsponse of PIF5 to light - And so on Or alternatively: - Response of WT and mutants to WL - Response of WT and mutants to Blue - Response to Red - And so on Also the discussion section is very dense. Could be helpful to make sub paragraphs here It could also be very informative if authors could make e.g. a PCA plot or something alike to visualize the gene expression and mRNA data with respect to light and with respect to mutant type. Also it would be informative to make a scheme depicting the results in a more visible way, showing based on the results what the sequence of events most probably is in response to different light colors. On the whole I think the data are interesting and it is a convincing story, but the MS is very hard to read/understand because it lacks a good structure.
Answer: We have completely redesigned the manuscript structure and introduced sub items where it was possible. They gave an additional description of miRNA targets.
We have made a graphical abstract in order to more fully characterize the observations made during the research.
The English text of the manuscript has been improved editing from American Journal Experts (Certificate verification key 1A0F-7257-F5BC-4C00-239F)
Reviewer 3 Report
Manuscript ‘Impact of light of different spectral composition on the morphology, photosynthesis, PIF protein content and expression of light-dependent microRNAs in Acrbidopsis thaliana pif mutant” reported PIF protein contents under three different light conditions (WL, BL and RL) from WT and pif mutants. However, better definitions of BL and RL light quantity and quality are needed.
Authors used 5 different pif mutants and compared their PIF contents and microRNA levels under three different light conditions. However, the data presentation might not be clear and concise toward the highlight and conclusion of this manuscript. The data could be more clearly presented to convey the appropriate scientific meaning.
The limited statement of results and conclusive remarks, which were poorly expressed.
Line102, given white light conditions in details. But missed the details of BL and RL light conditions (Line 106), such as dominant wavelengths of BL and RL? Without the details, experiments cannot be reproduced.
The Morphology section lacks narrative explanation. Authors should consider providing detailed similarity and differences between different light conditions and also on. E.g. the first sentence (line 206) reads not quite right. The reviewer believed there are differences among WT and pif mutants after reading Fig 1.
Authors claimed pif4, pif4pif5 and pif3pif4pif5 mutants … had a greater number of larger areas … (line 212-213), which was not agreed well as demonstrated in Fig 1. Fig 1 indicated WT has the largest leaves.
Authors may consider converting data (Tabular numbers) into ‘column graph’ (Fig 2 and Fig3) in order to indicate/compare/visualiee data easily.
There are 19 different microRNA presented in Fig 3. How to define them are ‘light-dependent microRNAs’.
What were the key discoveries from this research.
Were there relationships between changed transcriptional level of microRNA and photosynthesis activities?
Fig 4 showed positive western of PIF5 from pif5, pif3pif5, and pif3pif4pif5 mutants, Why? Similar question for presence of PIF3 in pif3 mutants? Only pif4 mutants supported by absence of PIF4 protein.
If mutants are partial, the results and conclusion may need to be reconsidered.
Author Response
1. Manuscript ‘Impact of light of different spectral composition on the morphology, photosynthesis, PIF protein content and expression of light-dependent microRNAs in Acrbidopsis thaliana pif mutant” reported PIF protein contents under three different light conditions (WL, BL and RL) from WT and pif However, better definitions of BL and RL light quantity and quality are needed.
Answer: We have added the spectra of light sources to the manuscript, now this is Figure 1.
2. Authors used 5 different pif mutants and compared their PIF contents and microRNA levels under three different light conditions. However, the data presentation might not be clear and concise toward the highlight and conclusion of this manuscript. The data could be more clearly presented to convey the appropriate scientific meaning.
Answer: We have revised the manuscript, we hope that in the modified form it is more understandable for the reader
3. The limited statement of results and conclusive remarks, which were poorly expressed.
Answer: We have revised the manuscript, we hope that in the modified form it is more understandable for the reader
4. Line102, given white light conditions in details. But missed the details of BL and RL light conditions (Line 106), such as dominant wavelengths of BL and RL? Without the details, experiments cannot be reproduced.
Answer: Done.
5. The Morphology section lacks narrative explanation. Authors should consider providing detailed similarity and differences between different light conditions and also on. E.g. the first sentence (line 206) reads not quite right. The reviewer believed there are differences among WT and pifmutants after reading Fig 1.
Answer: Done. The section completely redesigned
6. Authors claimed pif4, pif4pif5 and pif3pif4pif5 mutants … had a greater number of larger areas … (line 212-213), which was not agreed well as demonstrated in Fig 1. Fig 1 indicated WT has the largest leaves.
Answer: Section completely redesigned additional leaf area information has been included
7. Authors may consider converting data (Tabular numbers) into ‘column graph’ (Fig 2 and Fig3) in order to indicate/compare/visualiee data easily.
Answer: We understand the reviewer that the diagrams would make it easier to understand the text, however, such a presentation of the results by the team of co-authors was chosen due to the large amount of data presented in the article.
We had to reduce the number of presented results, leave only the most interesting ones, and embed the presentation into tables, otherwise it is very difficult to format in the text.
We tried to highlight the most key points in the tables with a black marker
8. There are 19 different microRNA presented in Fig 3. How to define them are ‘light-dependent microRNAs’.
Answer:We are grateful to the referee for pointing out the inaccuracy.
The fact is that these microRNAs in cis regions have been proven to bind to light-dependent transfactors, such as HY5 or phytochrome photoreceptors. We called them so on the basis of literature (doi:https://doi.org/10.3389/fpls.2018.00962 doi:https://doi.org/10.1186/s12864-017-3937-6 doi:https://doi.org/10.1371/journal.pgen.1007247).
However, in order not to mislead the reader, we have softened the wording throughout the texts.
9. What were the key discoveries from this research.
Answer: We are grateful to the referee for pointing out the inaccuracy.
We fixed the highlights and specifiers of the experiment results.
The conclusion is that we have clarified the list of microRNAs that can be regulated by PIF4 under blue light conditions, and which are most likely regulated by other transfactors.
10. Were there relationships between changed transcriptional level of microRNA and photosynthesis activities?
Answer: We are grateful to the reviewer for discovering an unfortunate inaccuracy in the manuscript. We did not found relationships between changed transcriptional level of microRNA and photosynthesis activities. Therfore we changed the last sentence in the Abstract. Therefore we changed the text in the Abstract." Unlike WT and other mutants, pif4 mutant responded to BL conditions not only by activating the expression of light-dependent miRNAs, but also by a significant increase in the expression of transcription factors and key light signaling genes. These molecular reactions do not affect the activity of photosynthesis, but may be involved in the formation of a light quality-dependent phenotype.".
11. Fig 4 showed positive western of PIF5 from pif5, pif3pif5, and pif3pif4pif5 mutants, Why? Similar question for presence of PIF3 in pif3 mutants? Only pif4 mutants supported by absence of PIF4 protein. If mutants are partial, the results and conclusion may need to be reconsidered.
Answer: We are grateful to the referee reviewer for carefully reading of the manuscript.
For western-blot-analysis we have used antibodies purchased from Agrisera (Sweden). For all anti-PIF very strong cross-reactivity was shown ( e.g. https://www.agrisera.com/en/artiklar/pif5-phytochrome-interacting-factor-5.html). We have tried different methods to, in first, to obtain enougth proteins to detect PIFs under light in adult plants, and, in second, to minimize such cross-reaction. But also, we cannot exclude, that multiple mutants can have their own properties in the early stages of their development.
We have shown that pif mutants remain such throughout ontogeny; in the case of PIF3, this cannot be said, and we added information about this to the manuscript.
The English text of the manuscript has been improved editing from American Journal Experts (Certificate verification key 1A0F-7257-F5BC-4C00-239F)
Round 2
Reviewer 1 Report
Comments on the revised version of the MS cells-2012863 submitted by Pashkovskiy et al.
This revised manuscript presents significant improvements relative to the previous version, but in my opinion, there are still improvements/clarifications needed before its publication. Again, the most important point to improve is the clarity of the “message”. The highlights are even worst than in the first version: “Light quality affects on the phenotype of pif mutants but not the photosynthetic rate”. But as shown in Table 1, there are large differences between the Pn values of the different plants grown under light conditions. As stated in li. 629-631, the authors indicate that light quality does not affect photosynthetic rate, which is misleading.
If I understand correctly, I suggest a title that could reflect the conclusion of this study.
Independent responses of photosynthesis and plant morphology to alterations of PIF proteins and light-dependent microRNA contents in Arabidopsis thaliana pif mutants grown under lights of different spectral compositions.
There are other details
1- Figure 1 presents the spectra of the different lights. The addition of pure blue and pure red lights is supposed to give magenta. Between 500 and 600 nm, there are no green nor yellow lights.
2- The results section begins (sub-section 3.1) with the effects of light spectra on the size and morphology of the leaf blades. But the values of the leaf area (=size) are presented in the 3.5 sub-section, in Table 1 whose title was not even adjusted.
3- The large differences between the ETR values estimated by chlorophyll fluorescence and the so-called Pn are still intriguing If the ETR are similar, then there are huge difference alternate electron sinks. The authors failed to mention that these measurements are quite different not only by the methods used, but more importantly, by the differences in light intensities. ETR and YII were measured under growth light intensities where gas exchanges were measured under saturating light. Therefore, it is important to use Pmax (or Amax) instead of Pn to emphasize this difference. This would help to explain, in part, the difference between ETR and CO2 assimilation measurements.
4- The text is much better, but still some improvements are needed. Please limit the use of the word variant for the different mutants. And not for the different light conditions (use light spectra). Also, the last paragraph of the Results section is awkward: “The maximum leaf area was in the WT in all light variants of light, and exceeded all other mutants and light variants”.
5- Finally, concerning the different effects of PIFs on the independent responses of photosynthesis and morphology to light spectra, it is relevant to mention that photosynthesis and growth can be affected differently by environmental factors (source-sink balance). For example, N and P limitations as well as water shortage are known to inhibit growth before affecting photosynthesis. Under source limitation (low light, low CO2, …), photosynthesis will limit growth.
I would consider the required improvements as minor for a first version, but for a revised version, the quality is disappointing and major revision is justified
Author Response
- Again, the most important point to improve is the clarity of the “message”. The highlights are even worst than in the first version: “Light quality affects on the phenotype of pif mutants but not the photosynthetic rate”. But as shown in Table 1, there are large differences between the Pn values of the different plants grown under light conditions. As stated in li. 629-631, the authors indicate that light quality does not affect photosynthetic rate, which is misleading.
Answer: We agree and improved this inaccuracy.
- If I understand correctly, I suggest a title that could reflect the conclusion of this study.
Independent responses of photosynthesis and plant morphology to alterations of PIF proteins and light-dependent microRNA contents in Arabidopsis thaliana pif mutants grown under lights of different spectral compositions.
Answer: We are grateful to the reviewer for formulating the title more fully reflecting the essence of the manuscript. We agree
- Figure 1 presents the spectra of the different lights. The addition of pure blue and pure red lights is supposed to give magenta. Between 500 and 600 nm, there are no green nor yellow lights.
Answer: We are grateful to the referee for pointing out the unfortunate mistake. We have re-determined the spectral composition of the light used by the light sources
- The results section begins (sub-section 3.1) with the effects of light spectra on the size and morphology of the leaf blades. But the values of the leaf area (=size) are presented in the 3.5 sub-section, in Table 1 whose title was not even adjusted.
Answer: Done
- The large differences between the ETR values estimated by chlorophyll fluorescence and the so-called Pn are still intriguing If the ETR are similar, then there are huge difference alternate electron sinks. The authors failed to mention that these measurements are quite different not only by the methods used, but more importantly, by the differences in light intensities. ETR and YII were measured under growth light intensities where gas exchanges were measured under saturating light. Therefore, it is important to use Pmax (or Amax) instead of Pn to emphasize this difference. This would help to explain, in part, the difference between ETR and CO2 assimilation measurements.
Answer: We agree and added to Discussion section the sentence "In addition, the measurements of Pmax were provided under light saturating conditions and Y(II) and ETR -under growth light intensity". In addition we replaced Pn with Pmax in corresponding sites.
- The text is much better, but still some improvements are needed. Please limit the use of the word variant for the different mutants. And not for the different light conditions (use light spectra). Also, the last paragraph of the Results section is awkward: “The maximum leaf area was in the WT in all light variants of light, and exceeded all other mutants and light variants”.
Answer: We are grateful to the reviewer for carefully reading the manuscript. We have tried to correct the text as much as possible.
- Finally, concerning the different effects of PIFs on the independent responses of photosynthesis and morphology to light spectra, it is relevant to mention that photosynthesis and growth can be affected differently by environmental factors (source-sink balance). For example, N and P limitations as well as water shortage are known to inhibit growth before affecting photosynthesis. Under source limitation (low light, low CO2, …), photosynthesis will limit growth.
Answer:
We are grateful to the reviewer for the made remark and agree with it.
This phrase added to text:
« It is known that the difference between the electron transport rate (ETR) or Y(II) and the net CO2 assimilation rate (Pmax) is an indicator of the existence of alternative electron sinks or as an example, nutrition limitations as well as water shortages [22,29]. We found the different effects of PIFs on the independent responses of photosynthesis and morphology to light spectra and did not observe a correlation between fluorescence parameters and Pmax, which is likely linked to the existence of alternatives to CO2 assimilation electron sinks. Parameters such as Y(II) and ETR vary greatly depending on the light and other conditions but the value of Fv/Fm is more conservative [22]. In addition, the measurements of Pmax were provided under light saturating conditions and Y(II) and ETR -under growth light intensity. »
English has been improved